# A systematic review and meta-analysis of the associations between interparental and sibling relationships: Positive or negative?

**Martina Zemp** [1]*, **Amos S. Friedrich** [1], **Jessica Schirl**[1], **Slava Dantchev** [1], **Martin Voracek** [2], **Ulrich S. Tran**[2]

**1** Department of Clinical and Health Psychology, Faculty of Psychology, University of Vienna, Vienna, Austria, **2** Department of Cognition, Emotion, and Methods in Psychology, Faculty of Psychology, University of Vienna, Vienna, Austria

* martina.zemp@univie.ac.at

**Data Availability Statement:** Data and syntax are available in OSF: https://osf.io/tbpgy/.

## Abstract

According to family systems theory, a family is regarded as an organized whole and relations within this system are interconnected. However, it is not clear to date whether the interparental and the sibling relationship are associated and, if such an association exists, whether it is positive or negative. Previous findings on the associations between the interparental and sibling relationships are inconsistent and there is as yet no pertinent review or meta-analysis. Therefore, this systematic review and meta-analysis set out (1) to aggregate previous studies investigating the links between the interparental and sibling relationships and (2) to examine potential moderators in this link. Based on 47 studies reporting 234 effect sizes ($N$ = 29,746 from six nations; 6–12 years; 49% boys), meta-analytic results suggest a small positive correlation between interparental and sibling relationship quality ($r$ = .14). Only the percentage of male children in the sample moderated this effect. Sex composition of sibling dyad and source of publication affected whether positive or negative associations were found. The findings support a growing consensus that family relations do not function in isolation, but are mutually interdependent, which should be considered in clinical practice.

## Introduction

The family is a crucial context for children's development and well-being, as it represents the primary place of socialization in a child's life. Family systems theory (FST) has received growing attention from developmental and clinical psychologists in the past few decades [1]. It argues for a comprehensive view that includes the various (interparental, parent-child, sibling) relationships in family dynamics. A family is regarded as an organized unit, and elements or subsystems within this system are inextricably interconnected and mutually interdependent; thus, dyadic interactions among family members may indirectly affect other members or dyads [2].

Particular consideration has been given to the interdependency between the interparental and the parent-child relationship in past research, which Erel and Burman [3] summarized in

**Funding:** The authors received no specific funding for this work.

**Competing interests:** The authors have declared that no competing interests exist.

a meta-analysis. The goal of the current meta-analysis is to apply this examination to the associations between two other family relations, namely, the quality of the interparental and the sibling relationship. Specifically, this study aims to examine the following objectives: First, are the interparental and sibling relationships significantly associated in a positive or negative direction? Second, do different moderators (i. a., operational definition and rater of the relationship quality, child age and sex, age difference between and sex composition of siblings, sibling order, family type, and other study characteristics) affect this association?

## The interparental relationship

Of the different family subsystems, the interparental relationship is regarded by many as key in ensuring family functioning and as the core of the family, versatilely shaping other family interactions [4]. The notion that the quality of the interparental relationship, i.e., the intimate relationship between the parents, is pivotal to children's well-being and that interparental conflict belongs to the strongest predictors of child maladjustment has been established throughout the past several decades [5].

The emotional security theory (EST) [6] holds that the quality of the interparental relationship affects children's psychological adjustment through their perceived emotional security, that is, their basic need of felt safety and security and confidence in their parents' abilities to preserve family stability. EST posits that maintaining emotional security in the family setting is a priority goal for children and insecurity is elevated in the face of interparental conflict. Drawing from attachment theory [7], EST shares a central attachment assumption; secure-base conceptions are central to both theories [8]. However, guided by principles of FST, EST differs from traditional attachment theory by placing an emphasis on the role of multiple family relationships in contributing to a child's emotional security. Thus, EST postulates that children also develop a sense of safety in terms of family relationships other than the parent-child relation, for instance, the interparental or the sibling relationship.

## The sibling relationship

Sibling relationships are another important cornerstone in a child's development for multiple reasons: Siblings often share a common and unique bond. The vast majority of children grow up with a sibling [9], and they spend a considerable amount of time together [10]. The sibling relationship is one of the most enduring relationships over the lifespan, and it plays an important role in children's socialization [11]. Siblings learn from each other during everyday moments of play and family activities and these interactions provide ongoing opportunities to acquire and improve social, emotional, and behavioral skills [12]. Siblings therefore have a strong mutual influence on how to manage their day-to-day life.

Along the lines of FST, it is imperative to consider these lifelong relationships in the context of other family relations, for example, the interparental relationship. Despite ample evidence that both family relationships are primary determinants of child developmental outcomes, relatively little headway has been made to systematically examine the links between the interparental and sibling relationships [13]. Furthermore, there is, to our knowledge, no systematic review or meta-analysis that has focused on these associations thus far. Therefore, it still remains unclear whether the interparental and the sibling relationship are associated and, if yes, whether the association is positive or negative.

## Positive associations between the interparental and sibling relationships

There are reasonable grounds to presume a positive association between the quality of the interparental relationship and the sibling relationship quality.

First, social learning theory suggests that destructive interparental conflict provides a model for dysfunctional sibling interactions [14]. By witnessing negative or aggressive interactions between parents, children may infer that this form of behavior is an acceptable way to resolve disagreements, and thus imitate these behaviors when interacting with their siblings [15]. Conversely, harmonious couples displaying constructive conflict model warm and functional interactions that may foster children's social adjustment [16].

Second, the interparental relationship can indirectly affect the sibling relationship through parenting. A large body of evidence supports the *spillover hypothesis*, which proposes that the quality of one relationship is positively linked to other relationships' quality within the family, due to a direct transfer of mood, affect, or behavior across family subsystems [3, 17]. According to this hypothesis, distressed couples become increasingly involved with their own relationship problems, depleting their resources necessary to rear their children sensitively, and are less emotionally available to adequately respond to their needs [18]. Parents reporting high conflict levels show less consistent and more dysfunctional parenting behavior [19]. Additionally, they are more likely to ignore negative sibling interactions or engage in differential treatment of children [20]. Parents' differential treatment means children perceive their parents behave differently toward themselves in comparison to their sibling, which can have negative consequences for child development according to prior research [21].

It is important to note, consistent with FST's central assumption of interdependency, that effects between family relations are assumed to be reciprocal, hence, may also travel from siblings to the parents' relationship. According to a meta-analysis [22], parents report lower satisfaction in their intimate relationships than do childless couples, and there is a negative link between the number of children and relationship satisfaction. Sibling conflict and concomitant negative affect impose additional strain for parents and can induce or exacerbate negative parental interactions [23].

A final factor that could underlie positive associations between the interparental and sibling relationships is the influence of a third family stressor that is neither part of the interparental nor the sibling relationship, such as one parent's stress at work, parental unemployment, or chronic illness of a family member. This process is referred to as *crossover*, a transfer of affect or behavior between people due to external stressors [23]. In other words, these stressors not only compromise the individual well-being of family members, but also form the basis for a strained family climate, thereby likely triggering conflict in different family relationships [24].

In sum, there are a number of reasons to expect that positive interparental relationship quality is associated with positive sibling relationship quality, and vice versa, negative interparental relationship quality comes along with negative sibling relationship quality in families.

## Negative associations between the interparental and sibling relationships

Some arguments speak in favor of a negative association between the interparental and the sibling relationship; thus, that low quality of the interparental relationship is associated with high sibling relationship quality, and vice versa.

First, the *compensation hypothesis* implies a negative association between the quality of two family relationships. As the opposite of spillover, compensation assumes that the transfer of affect within persons between relationships flows in a negative direction. Thus, compensation depicts a process in which family members seek opposite experiences in one relationship to compensate affection or balance deficiencies in another [3]. According to this hypothesis, parents experiencing interparental distress might want to compensate their negative couple interactions by positive parent-child interactions, which further lead to a positive relationship among siblings.

Second, a stress-buffering assumption suggests that children from families of high interparental conflict provide or seek support, help, and distraction in the sibling relationship [25], analogous to when faced with extra-familial stressors [26]. Hence, a strong tie between siblings may function as a buffer against the harmful effects of witnessing interparental discord or may assist with coping [27].

Third, warm and supportive sibling relationships may counterbalance for the lack of attention and security children receive from arguing or distant parents. EST offers a possible framework for this hypothesis, as it claims that children's felt emotional security in the family hinges on multiple family relationships [8], including the sibling relationship. Accordingly, brothers or sisters, as a proxy or surrogate for caregivers, may serve as a source of comfort and reassurance in times of high interparental conflict [21].

Fourth, it is also conceivable that children from harmonious couples tend to compete for their parents' attention through sibling quarrels and rivalry [1]. When interparental relationship quality is high, children may be viewed as intrusive or disruptive for the parental dyad, thereby creating tension in the sibling relationship. Alternatively, the couple relationship likely gets closer in the light of elevated sibling conflict, as they mobilize mutual support to efficiently cope with challenging sibling interactions [28].

Taken together, some considerations buttress the assumption that positive interparental relationship quality is associated with negative sibling relationship quality, and negative quality of the interparental relationship covaries with positive sibling relationship quality in families.

## The current meta-analysis

Systematic reviews and meta-analyses are an effective tool to quantitatively and systematically synthesize results on specific associations and to examine whether relevant factors moderate such associations. Therefore, the current study has two major goals: The first is to aggregate previous studies that have investigated the associations between the interparental and the sibling relationship quality. We determine the strength of the average correlation and its direction (whether the link is positive or negative).

The second goal is to examine whether the association between the interparental and sibling relationship quality is affected by potential moderators. Regardless of whether the association is positive, negative, or nonsignificant, it may be modulated by different factors. Thus, beyond investigating the strength and the direction of the link between the interparental and sibling relationships, identification of variables which moderate this association deserves attention. We examine 14 potential moderators (see Table 1), based on (a) theoretical considerations, (b) methodological considerations, (c) frequency of assessment in the empirical literature, and (d) previous suggestions of pertinent reviews and meta-analyses.

*[A] Operational definition of the interparental relationship quality*: Several dimensions to assess the quality of the interparental relationship have consistently emerged in the literature [6], which can be categorized in indicators of (1) positive dimensions (relationship satisfaction, adjustment, functioning, communication, interaction quality, affection, warmth, support, intimacy) or by indicators of (2) negative dimensions (interparental conflict, distress, discord, tension, aggression, hostility, violence). The operational definition must be controlled first by pooling the effect sizes to align the directions of the positive and negative dimensions of interparental and sibling relationship qualities (see below). Beyond, operational definition of the interparental relationship quality is considered as a potential moderator, since it probably affects the strength of association between the interparental and sibling relationships [3].

*[B] Rater of the interparental relationship quality*: The quality of the interparental relationship can be rated (1) by one or both parent(s), (2) by the child, or (3) by observer rating in the

**Table 1. Coding scheme of moderators.**

| Moderator | Categories of Coding |
|---|---|
| [A] Operational definition of the interparental relationship quality | (1) = Positive dimensions; (2) = Negative dimensions |
| [B] Rater of the interparental relationship quality | (1) = Parent(s); (2) = Index child; (3) = Observer; (4) = Unknown |
| [C] Operational definition of the sibling relationship quality | (1) = Positive dimensions; (2) = Negative dimensions |
| [D] Rater of the sibling relationship | (1) = Parent(s); (2) = Index child; (3) = Observer; (4) = Unknown |
| [E] Mean age of index children | (1) = 0–3 years; (2) = 4–6 years; (3) = 7–12 years; (4) = 13–18 years; (-99) = Unknown |
| [F] Mean age difference between siblings | In years (continuous); (-99) = Unknown |
| [G] Sex of children | Percentage male in child sample (continuous); (-99) = Unknown |
| [H] Sex composition of sibling dyad | (1) = Same-sex; (2) = Mixed; (3) = Unknown |
| [I] Sibling order of index children | (1) = Younger/est sibling; (2) = Elder/est sibling; (3) = Unknown |
| [J] Family type | (1) Cohabiting families, (2) Non-cohabiting families; (3) Mixed or unknown family types |
| [K] Sample type | (1) = Community; (2) = At-risk; (3) = Clinical; (4) = Unknown |
| [L] Type of study design | (1) = Cross-sectional design; (2) = Longitudinal design |
| [M] Level of statistical analysis | (1) = Within-subjects; (2) = Between-subjects; (3) = Unknown |
| [N] Source of publication | (1) = Peer-reviewed publications; (2) = Book chapters; (3) = Conference papers; (4) = Dissertations and Qualification theses; (5) = Unknown |

case of behavioral data, and the kind of assessment could affect the association between the interparental and sibling relationships.

*[C] Operational definition of the sibling relationship quality*: It has been established in the literature [29] that the quality of the sibling relationship is assessed by indicators of either (1) positive dimensions (relationship satisfaction, interaction quality, warmth, affection, support, companionship, closeness, cohesion) or (2) negative dimensions (sibling conflict, disagreements, quarrels, rivalry, fighting, aggression, hostility, tension, bullying). Analogous to the interparental relationship, the operational definition must be controlled first by pooling the effect sizes to align the directions of the positive and negative dimensions of interparental and sibling relationship qualities, and the operational definition of the sibling relationship quality is further considered as a moderator.

*[D] Rater of the sibling relationship quality*: Analogous to the interparental relationship, the quality of the sibling relationship can be rated (1) by one or both parent(s), (2) by the child, or (3) by observer rating in the case of behavioral data, and the method of assessment must be controlled.

*[E] Mean age of index children*: As the developmental stage influences the sibling relationship quality [11, 21] and the impact of the interparental relationship on children [15], studies are categorized depending on whether they focused on (1) infants (index children aged 0–3 years), (2) preschool children (4–6 years), (3) school-aged children (7–12 years), or (4) adolescents (13–18 years). Mean age of the index children is primarily considered to assign studies to the age groups. The index children are defined as the children who either reported on the sibling relationship quality themselves or for whom the parents or the observer rater reported on the sibling relationship quality. If characteristics of two (or more) siblings were reported, only data from the younger (or youngest) child are extracted for the sake of simplicity. If studies did not report the mean age, we consider the age range as the next step. In cases of broad age ranges overlapping with two or more of our selected age groups, we always use the lower

bound of the reported age range to classify the study (e.g., a sample with the age range of 5 to 11 years is assigned to the age group of preschool children).

*[F] Mean age difference between siblings*: Considering the previous finding that the age difference between siblings affects their relationship [21], we test the difference value in years, if reported, as a moderator.

*[G] Sex of children*: Given child sex has an impact on the sibling relationship quality [11] and on the effects of the interparental relationship on children [15], the percentage of male children in the whole sample, if reported, is tested as a moderator.

*[H] Sex composition of sibling dyad*: A meta-analysis of the impact of the sibling relationship quality on psychopathology of children and adolescents found that effects were moderated by sibling sex composition [21]. Therefore, we assess whether studies, if reported, examined (1) same-sex or (2) mixed sibling dyads.

*[I] Sibling order of index children*: To control for the possibility that the sibling order influences the quality of their relationship [25], we code whether the index children, if reported, were the (1) younger or youngest siblings or the (2) elder or eldest siblings.

*[J] Family type*: Family type, in particular the circumstance whether parents cohabit or are separated, affects the associations between different family relationships. Hence, we examine whether effects differ between (1) cohabiting families (including biological, adoptive, foster, and step-families), (2) non-cohabiting families (including separated and divorced families), and (3) mixed or unknown family types.

*[K] Sample type*: It is known that the quality of family relationships, their associations, and the individual well-being of family members differ between (1) community samples, (2) at-risk samples, and (3) clinical samples [3]. Studies with community samples do not focus on a particular subpopulation, thus, are based on non-stressed, healthy subjects. At-risk samples report elevated levels of stress or, respectively, deal with stressful life circumstances, such as below-average household income, parental unemployment, chronic illness of a family member, birth of a child with a physical disability, or presence of interparental violence. Clinical samples differ from at-risk samples in the level of psychopathology. In the present meta-analysis, studies are categorized as clinical samples if they either recruited participants in a clinical context (e.g., outpatient clinic, psychiatric consultation) or if any family member (parent, sibling) were diagnosed with a mental disorder.

*[L] Type of study design*: The strength and nature of the association between the interparental and sibling relationships might differ between (1) cross-sectional designs and (2) longitudinal designs [3]. We code (1) if the correlation coefficient reflects correlative associations at the same measurement time points and (2) if the correlation coefficient reflects prospective associations spanning across different measurement time points.

*[M] Level of statistical analysis*: Correlation coefficients further vary depending on whether effects are statistically computed on the (1) within-subjects level or on the (2) between-subjects level.

*[N] Source of publication*: It is generally recommended in meta-analyses [30] to control whether included studies are (1) peer-reviewed publications, (2) book chapters, (3) conference papers, or (4) dissertations and qualification theses.

## Method

### Search strategy

The systematic literature search followed PRISMA guidelines [31] (see S1 Checklist for the PRISMA checklist) and was conducted up to January 2020. No study protocol was preregistered for this systematic review and meta-analysis. The second and the third author conducted

the literature search and the study screening. We searched the following databases and search engines for relevant literature by using a combination of key search terms in English: Pubmed, PsycInfo, PSYNDEX, Scopus, and Web of Science. All search terms we used and an example of a full electronic search query are listed in S1 Table. The search terms underwent an iterative process of refinement after originally yielding nearly 100,000 search hits in a single database. We therefore excluded broad terms not specific to dyadic relationship descriptions (hereby excluded were the terms "child", "mother" or "maternal", and "father" or "paternal") and introduced proximity operators defining the maximum word distance between two terms to ensure relationship terms were close to dyadic terms. The distance parameter for the proximity search was chosen so that the extrapolated specificity of records dropped under 10%, as recommended by Mikolajewicz and Komarova [32]. During the process of literature search, references of previous reviews and studies were screened. In addition, literature already available within the research team as well as new literature emerging during the process of writing this paper was included as "other sources".

Fig 1 depicts the flow diagram of the literature search. The initial search yielded $k = 3,925$ records through databases and $k = 20$ through other sources. After removing duplicates ($k = 1,154$) using the *SRA DeDupe* tool [33], all titles and abstracts were screened in terms of our eligibility criteria. Among them, $k = 2,593$ studies were excluded because abstract analysis revealed that they were unsuitable for the current review (e.g., interparental relationship or sibling relationship not examined, not original empirical research, no quantitative data). The

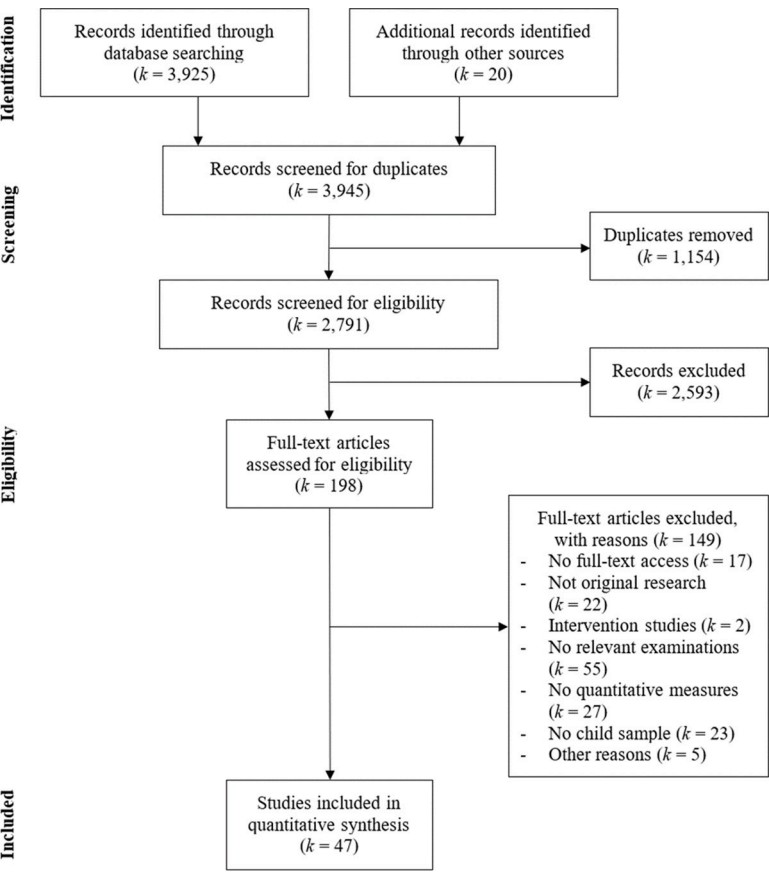

**Fig 1. PRISMA flow diagram of literature search.**

remainder ($k$ = 198), was given full consideration and after thorough full text analysis, a total of $k$ = 47 records met all eligibility criteria and were included in the meta-analysis [20, 25, 34–78]. Characteristics of the final set of studies ($k$ = 47) are listed in S2 Table.

## Criteria for inclusion and exclusion

To be included in the review, the studies had to meet the following criteria: (1) They had to examine the quality of the interparental and the sibling relationship (positive dimensions, such as relationship satisfaction, or negative dimensions, such as conflict or distress) in the same study, independent of family type or kinship (that is, biological, adoptive, foster, and step-parents as well as divorced or separated (non-cohabiting) parents were included). We exclusively focused on the quality of the interparental relationship, namely the intimate relationship between the parents, and thus excluded studies that only investigated the closely related, but distinct coparenting relationship (i.e., how parents cooperate and coordinate in childrearing and support each other in their parenting efforts). (2) At least one quantitative measure of the quality of each relationship (i.e., the interparental and the sibling relationship) was required, irrelevant of study design or method (e.g., self-reports, observational studies). (3) A calculation of any statistical parameter indicating the association between the two relationships had to be reported ($r$, $\beta$, $b$). (4) Statistical indices of sample size, means, and standard deviations ($N$, $M$, $SD$) of both relationships measures were obtainable. (5) The siblings had to be younger than 18 years, as this meta-analysis focuses on sibling relationships in childhood and adolescence.

Excluded were (1) theoretical papers and narrative reviews (not original empirical research), and (2) intervention studies involving treatments to enhance positive family (interparental or sibling) relationships. However, if these studies included pre-treatment assessments, control groups without treatment or a wait list control group, respectively, these specific data were considered for the present review as well.

## Data extraction

Studies were coded by two coders, the second and the third author. The coding procedure was done in a computer-based manner and was carried out according to established guidelines [79]. Thirty-four (69%) of the included 47 studies were independently double-coded by the two coders. The two coders achieved a Cohen kappa coefficient ($\kappa$) of .86. In the divergent cases (12.4%), the fourth author double-checked the values. Any remaining discrepancies were resolved through discussion among all coders and the first author and consensus could be reached. We extracted publication year, country of study, sample size ($N$), as well as relevant statistics ($M$, $SD$; in longitudinal designs from the first measurement point only) and key findings including correlation coefficients ($r$, $d$, $\beta$, $OR$) for the interparental and the sibling relationship quality indexes. In addition, we extracted information on potential moderators according to our coding scheme listed in Table 1. Coded moderators of the final set of studies ($k$ = 47) are listed in S3 Table.

## Description of study sample

The $k$ = 47 studies amounted to a combined sample size of $N$ = 29,746 overall, individually ranging from $n$ = 25 to $n$ = 8,122. The studies reported a total of 234 relevant effect sizes, of which 188 (80%) indicated positive associations, 40 (17%) negative associations, and 6 effect sizes (3%) no association. Mean age of child participants predominantly lay between 6 and 12 years, and the percentage of male children across all samples averaged 49% (range = 0%–100%). The vast majority of studies were conducted in the United States ($k$ = 38), other countries included the United Kingdom ($k$ = 3), Canada ($k$ = 2), Netherlands ($k$ = 2), Israel ($k$ = 1),

and Taiwan ($k$ = 1). With $k$ = 42 (89%), most studies referred to community samples (healthy subjects without known risk), while $k$ = 4 (9%) investigated at-risk samples (e.g., low-income families, families exposed to community violence, families with intimate partner violence). One study (2%) examined all three population categories–community, at-risk, and clinical–separately in subgroups within the same investigation (i.e., children diagnosed with current major depression with a depressed parent, depressed children without a depressed parent, children considered at high-risk for depression, and children considered at low-risk for depression (normal controls); see Weaver-Graham [77]). Publications mostly stemmed from peer-reviewed journals (68%) while the remaining studies were dissertation theses (32%). Articles were published between 1978 and 2020. A detailed list of included studies can be found in S2 Table.

## Methodological quality of studies

To be included in the meta-analysis, all studies had to include at least one quantitative measure to assess the quality of both the interparental and the sibling relationship, irrelevant of study design or method (e.g., self-reports, observational studies). It is a common shortcoming in the family science literature that child outcomes are examined through parental reports, although it is more important what children perceive that their families do, not what parents think children perceive. Even stronger data validity is provided by combinations of multiple raters or sources and multiple methods. Of the included studies in the current meta-analysis, $k$ = 32 (68%) reported parent ratings of the interparental relationship quality, $k$ = 8 (17%) reported child ratings, $k$ = 2 (4%) reported observer ratings, and $k$ = 5 (11%) included multiple sources. For assessing the sibling relationship quality, $k$ = 9 (19%) reported parent ratings, $k$ = 25 (53%) reported child ratings, $k$ = 7 (15%) reported observer ratings, and $k$ = 6 (13%) included ratings from multiple sources. Thus, the proportion of studies reporting child ratings or multiple sources is considerable.

The reliabilities of the measures used to assess the relationship quality were acceptable overall: Observational studies including multiple coders all reported percent agreements ≥ 70%. The lowest Cronbach α coefficients reported for the interparental relationship were ≥ .65 in 72% ($k$ = 34) of included studies and < .65 in one study only; 15% ($k$ = 7) did not report on internal consistency. The lowest Cronbach α values reported for the sibling relationship were ≥ .65 in 77% ($k$ = 36) of included studies and < .65 in three studies; 17% ($k$ = 8) did not report on this statistic. As the few low α values are most likely due to the inherent inaccuracies of young children's reports, and all constructs included measures with α > .70, it is reasonable to generally assume adequate internal consistency of the data reported in the primary studies.

Large sample sizes reduce the risk of inadequate data representation and improve statistical power. Twenty-five studies (53%) reported sample sizes of $N$ < 100, 12 (26%) of $N$ between 100 and 300, and 10 (21%) of $N$ > 300. Remarkably, the majority of sample sizes included different family members (parents, children, siblings), such that a sample of 100 families in fact reflected more than 100 individuals. Hence, the included studies examined large samples overall. Furthermore, studies were weighted according to their sample size in the present meta-analysis.

Of the included studies, $k$ = 38 (80%) were cross-sectional and $k$ = 9 (20%) implemented a longitudinal design. An inherent advantage of longitudinal designs is that they establish a temporal–and thus more plausibly causal–connection between constructs. One fifth of the included studies provide such a longitudinal perspective and for this reason are given separate consideration in our moderator analysis.

By virtue of our eligibility criteria, theoretical papers and narrative reviews (not presenting original research), and intervention studies involving treatments to enhance family

relationships (unless they included pre-treatment assessments or data of control groups without treatment), were excluded from this meta-analysis. The process of peer-reviewed publishing is one of the principal means to ensure high scientific quality, and thus an important indicator for quality assessment. Out of the included studies, $k = 32$ (68%) were published in peer-reviewed journals; the remaining $k = 15$ (32%) studies were all dissertation theses. These too undergo scrutiny and approval by colleagues, though arguably not by the same standards as in peer-reviewed journals.

Taken together, our rigorous criteria applied to include or exclude studies in the search strategy warrant high methodological quality of primary studies using reliable and valid measures of the core constructs. This is in accordance with our systematic quality assessment of primary studies using the Mixed Methods Appraisal Tool [80] (see S4 Table).

## Data syntheses

All reported associations between the interparental and sibling relationship qualities were converted into the effect-size metric $r$, using formulae provided in Cooper et al. [81] (chapter 11). Effect sizes were pooled in such a way that the sign of the effect size indicated positive or negative associations. This entailed the switching of the reported sign of all effects, for which the dimensions of interparental and sibling relationship qualities differed in direction.

As many studies reported more than one effect size (e.g., because they presented associations in more than one sample, associations of more than one scale or with multiple subscales of one measure, or associations at different time points) three-level meta-analytic models [e.g. 82] were used in analysis. Three-level meta-analysis is similar to conventional meta-analysis, but allows including more than one effect size per study by treating the data structure as hierarchical, just like conventional multilevel models do.

Three-level meta-analytic models were fitted on the data, treating effect sizes (level 1) as nested within studies (level 2: within-study level). Level 3 constituted the between-study level. The models estimated variance components within studies ($\sigma_2$) and between studies ($\sigma_3$). We report square roots of the variance components (i.e., $\sigma = \sqrt{\sigma^2}$), as this allows for higher precision with fewer digits and places estimates directly on the scale of the effect sizes themselves. First, a baseline model was fitted on the data, which estimated the correlation between the interparental and sibling relationship quality across all studies and effect sizes. We then subsequently tested for the effects of moderating variables, using one moderator at a time. In a final step, all significant ($p < .05$) moderators were investigated in a combined model. In addition to this moderator analysis, we also examined with logistic regression analysis whether any of the moderating variables predicted the sign of the correlation between the two relationships, i.e., specifically whether correlations indicated positive associations (positive sign) or negative associations (negative sign).

For the investigation of publication bias [e.g. 83], we examined on the one hand effects of publication source (moderator [N]) and on the other hand whether smaller, and, hence, less precise, studies reported stronger effects than larger, and more precise, studies (i.e., small-study effects). For this, a further moderator analysis was conducted, using the inverse of the effect-size standard error (as a proxy of precision) as a moderator.

For all computations, *metafor* in R [84] was used. Maximum likelihood was utilized for parameter estimation. Standard errors were estimated with robust methods [85], tests of significance were based on the Knapp-Hartung method [86]. For the variance components, 95% profile-likelihood confidence intervals (CIs) [e.g. 87] are reported. Concerning effect-size heterogeneity, we present $Q$ tests and $I^2$ values (overall and partitioned to all variance components, using the formulae available on http://www.metafor-project.org/doku.php/tips:i2_

multilevel_multivariate). Overall $I^2$ values of ~25%, ~50%, and ~75% were interpreted to indicate low, medium, and high excess heterogeneity. For the moderator analyses, we present the results of robust omnibus $F$ tests of significance [85] and variance explained ($R^2$) of overall and partitioned excess heterogeneity. Following recommendations in the literature [e.g. 81], $r$ values were not converted into Fisher $z$ values for analysis, as variance components cannot be easily converted between these two metrics.

## Results

### Baseline model

Overall, there were 234 effect sizes reported in the 47 studies ($Mdn$ = 3, min = 1, max = 24). Across all studies and effect sizes, the average correlation between the interparental and sibling relationship quality amounted to $r$ = .14, $SE$ = 0.02, 95% $CI$ [0.10, 0.18], $p < .001$. Overall, effect-size heterogeneity was high, $Q(233)$ = 4564.92, $p < .001$, $I_{overall}^2$ = 92.26%. Heterogeneity was similar on level 2 (within studies) and level 3 (between studies), $\sigma_2$ = 0.10, 95% $CI$ [0.09, 0.12], $I_2^2$ = 41.20%, and $\sigma_3$ = 0.11, 95% $CI$ [0.08, 0.15], $I_3^2$ = 51.05%, respectively. We controlled for multiple effect sizes within the same study, but did not control for multiple publications by the same author. There were in total eight studies that were from the same four first authors (each of those having published two studies). Including this information on (identical) authorship as a further level in the model (level 4) resulted in an estimate of the square root of this variance component of 0 (i.e., no variance was attributable to authorship). Using authorship instead of study as level 3 in the model resulted in the same estimates (to the second digit).

### Moderator analyses

The results of the moderator analyses are listed in Table 2. For sample type [K], effect sizes from at-risk and clinical samples were combined, because there was only one effect size from a clinical sample [77]. Level of statistical analysis (within-subjects vs. between-subjects; moderator [M]) was not tested, as it exhibited no variability in the data. For source of publication [N], only peer-reviewed publications and dissertations were contrasted, as there were no other publication types in the data. Mostly, the candidate moderator variables did not affect correlations to a relevant extent. The large deviation concerning the 'unknown' category in moderator [D], namely the rater of the sibling relationship, could be traced to a single study [54] and, hence, is likely of no relevance overall. Only sex of the index children [G] was a relevant moderator: the correlation between the interparental and sibling relationship quality was relatively stronger among studies with fewer male index children. Publication type [N] narrowly missed nominal significance ($p$ = .07). No combined model was tested, as only one moderator had proved significant.

Using logistic regression analysis and a stepwise backward approach, we examined whether the moderators predicted the sign of the correlations (positive vs. negative). The first step included all moderators, except mean age difference between siblings [F], because of its many missing values, and level of statistical analysis [M], because it showed no variability in the data. The final model retained moderators sex composition of sibling dyad [H] and source of publication [N] and explained 11% of the variance (Nagelkerke $R^2$), $\chi^2$ = 13.77, $df$ = 3, $p$ = .003. Evidence of negative associations (negative sign of the correlation) was more often reported in studies with same-sex sibling dyads [H] than in studies with mixed ($OR$ = 0.24, 95% $CI$ [0.59, 0.95], $p$ = .042) or unknown dyads ($OR$ = 0.18, 95% $CI$ [0.41, 0.76], $p$ = .020), and more often in dissertations than in peer-reviewed publications ([N]; $OR$ = 3.99, 95% $CI$ [1.69, 9.43], $p$ = .002).

**Table 2. Results of the moderator analyses.**

| Moderator | m | Estimate (SE) | 95% CI | Robust F test | R² (overall / lvl 2 / lvl 3) |
|---|---|---|---|---|---|
| [A] Operational definition of the interparental relationship quality | | | | $F(1, 45) = 0.50$ | 0% / 0% / 1% |
| Positive dimensions | 119 | 0.13 (0.02)*** | [0.08, 0.18] | | |
| Negative dimensions | 115 | 0.15 (0.02)*** | [0.10, 0.19] | | |
| [B] Rater of the interparental relationship quality | | | | $F(3, 43) = 0.96$ | 3% / 1% / 4% |
| Parent(s) | 176 | 0.13 (0.02)*** | [0.09, 0.17] | | |
| Index child | 41 | 0.20 (0.05)*** | [0.10, 0.30] | | |
| Observer | 10 | 0.08 (0.09) | [-0.10, 0.26] | | |
| Parents & observer mixed | 7 | 0.08 (0.12) | [-0.15, 0.31] | | |
| [C] Operational definition of the sibling relationship quality | | | | $F(1, 45) = 1.36$ | 1% / 2% / 0% |
| Positive dimensions | 115 | 0.12 (0.02)*** | [0.08, 0.17] | | |
| Negative dimensions | 119 | 0.15 (0.03)*** | [0.10, 0.21] | | |
| [D] Rater of the sibling relationship | | | | $F(3, 230) = 2.00$ | 2% / 3% / 1% |
| Parent(s) | 50 | 0.19 (0.04)*** | [0.11, 0.28] | | |
| Index child | 138 | 0.12 (0.03)*** | [0.07, 0.17] | | |
| Observer | 43 | 0.16 (0.04)** | [0.07, 0.25] | | |
| Unknown | 3 | -0.09 (0.14)[a] | [-0.37, 0.20][a] | | |
| [E] Mean age of index children | | | | $F(3, 43) = 0.12$ | 1% / 0% / 1% |
| 0–3 years | 25 | 0.12 (0.05)* | [0.01, 0.23] | | |
| 4–6 years | 38 | 0.15 (0.04)** | [0.06, 0.24] | | |
| 7–12 years | 123 | 0.14 (0.02)*** | [0.09, 0.19] | | |
| 13–18 years | 48 | 0.14 (0.05)** | [0.04, 0.23] | | |
| [F] Mean age difference between siblings | 93 | -0.01 (0.04) | [-0.08, 0.07] | $F(1, 19) = 0.06$ | 0% / 0% / 0% |
| [G] Sex of children (% male) | 207 | -0.0015 (0.0003)*** | [-0.0021, -0.0008] | $F(1, 40) = 22.29$*** | 1% / 1% / 1% |
| [H] Sex composition of sibling dyad | | | | $F(2, 44) = 0.81$ | 3% / 0% / 6% |
| Same-sex | 18 | 0.10 (0.06) | [-0.02, 0.23] | | |
| Mixed | 144 | 0.12 (0.02)*** | [0.07, 0.17] | | |
| Unknown | 72 | 0.18 (0.04)*** | [0.10, 0.27] | | |
| [I] Sibling order of index children | | | | $F(2, 44) = 0.37$ | 1% / 0% / 3% |
| Younger/est sibling | 79 | 0.12 (0.04)** | [0.05, 0.19] | | |
| Elder/est sibling | 26 | 0.16 (0.05)** | [0.05, 0.26] | | |
| Unknown | 129 | 0.15 (0.03)*** | [0.09, 0.20] | | |
| [J] Family type | | | | $F(2, 44) = 0.74$ | 10% / 0% / 18% |
| Cohabiting families | 167 | 0.13 (0.02)*** | [0.09, 0.18] | | |
| Non-cohabiting families | 6 | 0.28 (0.12)* | [0.04, 0.53] | | |
| Mixed or unknown family types | 61 | 0.13 (0.04)** | [0.05, 0.21] | | |
| [K] Sample type | | | | $F(1, 45) = 0.93$ | 4% / 0% / 8% |
| Community | 226 | 0.13 (0.02)*** | [0.09, 0.17] | | |
| At risk (m = 7) & clinical (m = 1) | 8 | 0.23 (0.10)* | [0.03, 0.43] | | |
| [L] Type of study design | | | | $F(1, 45) = 3.40$ | 2% / 1% / 3% |
| Cross-sectional | 183 | 0.15 (0.02)*** | [0.10, 0.19] | | |
| Longitudinal | 51 | 0.11 (0.03)*** | [0.06, 0.17] | | |
| [N] Source of publication | | | | $F(1, 45) = 3.40$ | 5% / 0% / 10% |
| Peer-reviewed publications | 151 | 0.16 (0.03)*** | [0.11, 0.21] | | |

*(Continued)*

**Table 2.** (*Continued*)

| Moderator | m | Estimate (SE) | 95% CI | Robust F test | R² (overall / lvl 2 / lvl 3) |
|---|---|---|---|---|---|
| Dissertations | 83 | 0.09 (0.03)** | [0.03, 0.15] | | |

*Note. m* = number of effect sizes, *CI* = confidence interval. Estimates are average correlations in the designated categories or slopes (in the case of moderators [F] and [G]).

[a]Robust estimates of the *SE* and *CI* are not provided for this category, and no robust omnibus test is provided for the moderator itself, as all effect sizes stemmed from only one study (Liu, 2006); single-study clusters are problematic for the robust estimator, which exploits the cluster structure of the data. Moderator [M], i.e. level of statistical analysis, was not tested as it exhibited no variability in the data.

*p < .05

**p < .01

***p < .001.

Effect sizes were not associated with their inverse standard errors, *B* = 0.002, *SE* = 0.002, 95% *CI* [-0.003, 0.007], *p* = .49. Hence, there was no indication of small-study effects in this meta-analysis. A funnel plot of all effect sizes is presented in Fig 2. Some effect sizes were visibly oriented along curved lines in this plot. This was caused by clusters of effect sizes, which stemmed from the same studies (e.g., the ostentatious line at the top of the funnel plot extending to the right shows the 24 effect sizes of Haj-Yahia and Abdo-Kaloti [47]). Other than this, there was no obvious visual indication of effect-size asymmetry. Yet, the above result on evidence of negative correlations being more likely for dissertations may indicate some bias in peer-reviewed publications nonetheless.

## Discussion

When viewing the family unit as an organized system, each subsystem differs depending on the individuals' personalities and characteristics of the respective relationship. However,

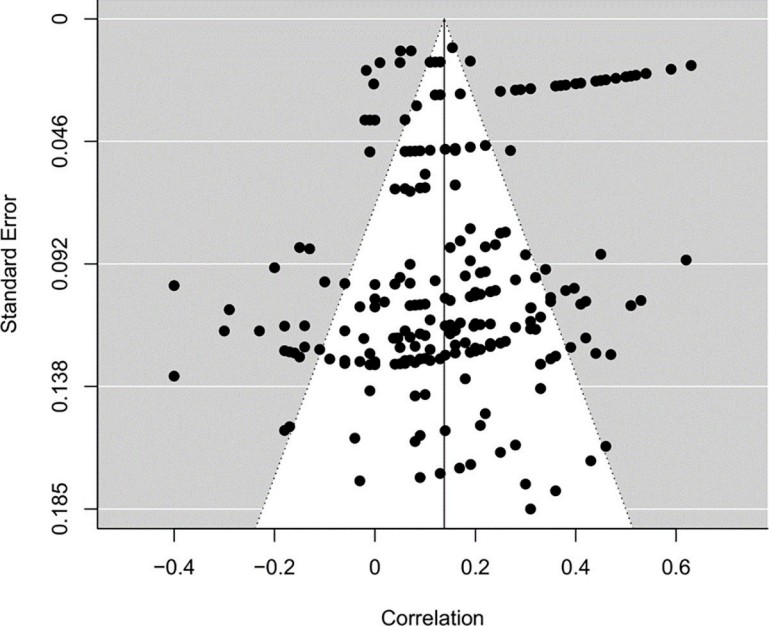

**Fig 2. Funnel plot.**

despite these distinctions, each family relationship works together to create an intertwined, whole unit being more than the sum of its parts [2]. Two family relationships, the interparental and the sibling relations, are primary determinants of child developmental outcomes. This meta-analysis quantified the empirical research on the associations between the quality of the interparental and the sibling relationship.

Across all studies and effect sizes, the average correlation between the interparental and sibling relationship quality reflected a small, but significant, positive association. Several theoretical frameworks explain why it is reasonable to expect that a high interparental relationship quality often comes along with positive sibling relationships in families, and vice versa. One the one hand, social learning theory [14] suggests that both positive and negative parental interactions elicit or exacerbate behavior in siblings that is similar to what they witness in their parents' interactions [16, 17]. On the other hand, according to the spillover hypothesis, the interparental relationship can influence the sibling relationship indirectly through parenting or coparenting, respectively [3, 17]. For instance, interparental conflict can disrupt childrearing practices or interfere with sensitive parenting [18]. A meta-analysis found a significant association between interparental conflict and dysfunctional parenting behaviors, with strongest effect sizes regarding harsh discipline and parental acceptance [19]. Similar to the present work, a recent study [88] examined spillover and compensation processes in adult relationships with intimate partners and siblings. The results supported the spillover hypothesis in general, as individuals with more self-disclosure or conflict with their partners also tended to report more self-disclosure or conflict with their siblings. However, the present meta-analysis differentiates from that study and much previous research [3] in that we examined two family relationships where no family member is part of both relationships (e.g., parents and siblings instead of parents and parent-child relations). It is evident that affect contagion more likely occurs from one family relationship to the other if one person is involved in both relationships (e.g., interparental conflict is posited to erode parents' mood, and this in turn disrupts coparenting, parenting, or parent-child interactions). This is according to the original framework of the spillover hypothesis postulating that affect within persons transfers directly from one relationship to another [3]. Thus, it is about the same person in different relationships, which was not investigated in the present meta-analysis. This means that the present data, as (co)parenting was not examined, do not reveal whether spillover of affect within parents underlies our findings. This is an important limitation and we may assume that the association between coparental alliance, parenting behavior, or parent-child interactions and sibling relationship quality would have yielded stronger effects than the association between interparental and sibling relationships. Future research needs to examine those postulated mechanisms.

However, it can be assumed that affect contagion also occurs from one family relationship to the other if no person is involved in both relationships. For example, as the experience of seeing or hearing displays of anger between parents is itself aversive to children, repeated exposure to interparental hostility takes a direct toll on children. When faced with interparental conflict, children react emotionally (e.g., fear, distress), behaviorally (e.g., intervening, avoidance), physiologically (e.g., skin conductance level reactivity), and cognitively (e.g., insecure representations, self-blaming) [89]. Children's reactivity to marital conflict is, according to the EST [6], an expression of perceived threat to their sense of security. Early experimental studies have demonstrated that children's exposure to adult background anger, i.e. an angry interaction of two actors in the background of the experimental room, increased their behavioral and emotional distress and heightened aggression toward peers [90, 91]. Similarly, exposure to videotaped couple arguments increased the likelihood of aggressive behavior in children [92]. From that perspective, a direct association between interparental conflict and sibling aggressive interactions, not mediated by parenting, is very plausible.

Another theoretical implication of this study relates to our impression that scholars often use different labels and operational definitions for similar constructs in this field (e.g., relationship satisfaction / quality / adjustment / functioning for positive dimensions or conflict / distress / discord / disagreements for negative dimensions of relationship measures). Inconsistence in the terminology and confusion in the conceptual operationalization of family relationships constitute a major challenge particularly for meta-analyses, as it is rarely possible to create an exhaustive list of search terms. We tried to avoid potential pitfalls by a prudent coding of the operational definitions of the different dimensions of relationship quality. Although relationship science is complex and manifold and there are no established international definitions, the field would generally benefit from greater conceptual consensus. We recommend that empirical studies at least mention frequently used global key words in their articles (e.g., relationship quality) if they focus on one narrow indicator of it (e.g., intimacy).

Moderation analyses revealed that only one factor, namely sex of children, significantly affected the strength of the association between the interparental and sibling relationship quality. Specifically, the correlation was stronger among studies with fewer male children (smaller percentage of boys in the sample). One potential explanation of this finding can provide developmental models of gender socialization: Girls are commonly socialized to value interdependence and connectedness in close relationships while boys are often supported to develop greater independence and autonomy [93]. As a result, boys are hypothesized to develop greater concern for self-protection, whereas girls experience pressure to conform to communal gender roles that are manifested in greater interest toward communion and the merging of individuals within social systems [94]. Prior research has found that they are particularly likely to ruminate about and intervene in family interactions, exhibit greater sensitivity to interparental disagreements, and report more self-blaming appraisals in the face of interparental conflict [95, 96]. Hence, gender differences are plausible in this meta-analysis, with girls being more likely to incorporate interpersonal behavior from their parents into their sibling interactions than boys. In other words, girls might be more susceptible to imitate relationship interactions leading to greater associations between the interparental and sibling relationship in families with daughters. That said, it must be taken into account that the majority of studies reporting on this variable reported a percentage of males lying between 30% to 60% (that is, only very few investigated samples with none or few [<30%] or many [>60%] boys). Future studies with more variation on the child sex variable, or among girls or boys exclusively, are important to inform about potentially distinct processes in those families.

Of all effect sizes, 17% indicated negative associations between the interparental and sibling relationship quality. Why in some cases are siblings able to maintain supportive relationships that buffer against a negative interparental relationship quality, or conversely, why can parents in some families preserve high intimate relationship quality in the face of a negative sibling relationship? In our data, there was evidence that two moderators affected whether positive or negative associations were found: the sex composition of sibling dyad and the source of publication. Negative links were more often reported in studies with same-sex sibling dyads and more often in dissertations than in peer-reviewed publications. It is possible that siblings' personalities and the match between them, such as among same-sex siblings, or a history of supportive sibling relationships especially in times of stress may enable children to turn to siblings for support when coping with interparental conflict. Two sisters or two brothers, respectively, may mutually serve as protective figures to a greater extent than mixed sibling dyads. By a greater sensitivity and by acting as role models, siblings may assist particularly same-sex siblings to handle interparental relationship distress in everyday family life. In a similar vein, the sex composition of siblings has emerged as important factor in previous literature documenting that sisters provide more comfort to their siblings, particularly to other sisters [26].

However, one must bear in mind that only $k = 4$ studies examined same-sex sibling dyads and this circumstance could have affected our results. On the other hand, some couples are particularly robust against sibling rivalry or bullying because they dispose of specific stress resistance resources, such as successful constructive communication, successful coparenting, and dyadic coping (i.e., how partners support each other to manage stress as a couple) [97]. We can only speculate–and future studies should examine–whether parents of same-sex siblings score higher on these resources in general such that they are better capable to maintain a good intimate relationship functioning in spite of negative sibling relationships. Moreover, given the above result on evidence of negative associations being more likely for dissertations than in peer-reviewed publications, we cannot rule out that a potential publication bias played a role in this finding.

## Practical implications

The meta-analytic evidence pointing towards a positive link between interparental and sibling relationship quality calls for several practical implications. First, the findings of this study warrant preventative measures building on psychoeducation and parental training aimed at highlighting the interdependence of family relationships. It is crucial to raise parental awareness about the importance of positive interparental interactions in promoting healthy sibling relationships. Parents act as central models of social relationships within the family. Supporting parents in strengthening their interpersonal bond can help them set a positive example and equip their children with the necessary skills to foster their own positive relationships. If we assume that the interparental relationship spills over into the sibling relationship, one route that could target this, would be to primarily strengthen the interparental relationship. A randomized controlled trial has shown that a couple-focused intervention can reduce child behavioral problems, mediated by enhanced relationship quality in mothers and improved parenting behavior in fathers [98]. While the importance of the enhancement of the interparental relationship to reduce and prevent child behavior problems is increasingly recognized [99], the effects of couple-oriented programs on the sibling relationship have been largely neglected so far.

However, the effects may also occur from the sibling domain onto the parental bond. Negative sibling relationships may act as significant family stressors, placing a heavy strain on the interparental relationship [23]. Thus, programs that directly target the sibling relationship may be an alternative route. Employing tailored family interventions aimed at managing sibling conflict may therefore not only improve the sibling relationship, but also relief interparental distress, thereby enhancing the entire family system. Programs that address sibling conflict and aggression are scarce, however there is a handful of studies that have put forward interventions that are aimed at either improving children's social skills or at teaching parents mediation techniques in the face of sibling conflict [100–102].

While targeting either the interparental or sibling domain may help improve family relationships, it may be worth for future work to embrace a more integrated family systems approach that holds promise to influence multiple family subsystems simultaneously. Couple-focused interventions including both components of interparental bonds and parenting or coparenting have been successful at reducing child behavior problems by improving the interparental relationship quality first and in turn the parent-child relationship [103]. Parenting-oriented programs, such as Triple P [104], on the other hand may hold promise to indirectly influence both the interparental and the sibling domain. Previous studies have found that variants of the Triple P training have the potential to enhance the interparental relationship quality [105], as well as to reduce sibling conflict [106].

Intervention strategies primarily focusing on reducing problem behavior, albeit alleviating negative behavior, may not equip family members with the necessary resources to sustain positive family relationships in the long run. The sibling literature has previously made efforts to highlight the importance of promoting prosocial sibling engagement and conflict management strategies as opposed to focusing on the reduction of sibling conflict exclusively [107]. Thus, in the light of the present findings it may be useful to develop a similar approach in order to promote a spillover of positive family interactions and reinforce healthy patterns instead.

### Limitations and future directions

A number of limitations of the present meta-analysis merit consideration. First, the average correlation between the interparental relationship quality and the sibling relationship quality nominally was significant, but small with regards to the magnitude of the effect. Implications in terms of practical significance should thus be interpreted with caution. Second, almost all studies were conducted in the US or other Western, well-educated, democratic, and individualistic countries, which limits the generalizability of the results especially to other cultures and less developed countries. Third, the majority (80%) of the studies implemented a cross-sectional design; therefore, causal inferences cannot be drawn. Hence, it is not possible to conclude whether the interparental relationship affected the sibling relationship or the other way around. In addition, all primary studies were based on a between-subject design. To expand the understanding of the association between the interparental relationship and the sibling relationship *within* families, future research should examine within-subjects fluctuations as well, using intensive longitudinal methods (e.g., in form of diary studies). Observational methods are also essential to understand family processes and should be used more often in the future. Last, for reasons of simplicity, only data from the younger (or youngest) child were extracted if characteristics of two (or more) siblings were reported. This decision has some disadvantages, as older siblings are more likely to be the leaders, superiors, and protectors in the sibling dyad. Hence, we cannot preclude the possibility that the pattern of results would have looked different when examining the older siblings' perception of the sibling relationship quality.

### Conclusion

To our knowledge, this is the first meta-analysis to integrate the existing findings on the association between the interparental and the sibling relationship quality. The results suggest a small positive correlation between the two family relationships. Our study underlines the importance of taking on a family systems perspective when considering the implementation of family-based interventions tailored to improve family interactions. It is therefore deemed important to place an emphasis on enhancing positive interactions across multiple family subsystems, including the parental, parent-child and sibling subsystems, as these may map onto one another and simultaneously build a positive cascade promoting sustained family positivity, well-being, and healthy development.

### Supporting information

**S1 Checklist. PRISMA checklist.**
(DOC)

**S1 Table. Key search terms for literature research.**
(DOCX)

**S2 Table. Study characteristics of included studies ($k = 47$).**
(DOCX)

**S3 Table. Coded moderators of included studies ($k = 47$).**
(DOCX)

**S4 Table. Quality assessment of included studies ($k = 47$).**
(DOCX)

## Acknowledgments

We thank Carmen Schneckenreiter for assisting with the quality assessment of the primary studies.

## Author Contributions

**Conceptualization:** Martina Zemp, Martin Voracek.

**Data curation:** Amos S. Friedrich.

**Formal analysis:** Ulrich S. Tran.

**Methodology:** Martina Zemp, Amos S. Friedrich, Jessica Schirl.

**Software:** Ulrich S. Tran.

**Supervision:** Martina Zemp.

**Visualization:** Ulrich S. Tran.

**Writing – original draft:** Martina Zemp.

**Writing – review & editing:** Amos S. Friedrich, Slava Dantchev, Martin Voracek.

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
