## [Decision Letter · Decision Letter 0]

24 Jun 2021

PONE-D-21-07364

A systematic review and meta-analysis of the associations between interparental and sibling relationships: Spillover or compensation?

PLOS ONE

Dear Dr. Zemp,

Thank you for submitting your manuscript to PLOS ONE. After careful consideration, we feel that it has merit but does not fully meet PLOS ONE’s publication criteria as it currently stands. Therefore, we invite you to submit a revised version of the manuscript that addresses the points raised during the review process.

We look forward to receiving your revised manuscript.

Kind regards,

Livio Provenzi

Academic Editor

PLOS ONE

Journal Requirements:

2. We note that this manuscript is a systematic review or meta-analysis; our author guidelines therefore require that you use PRISMA guidance to help improve reporting quality of this type of study. Please upload copies of the completed PRISMA checklist as Supporting Information with a file name “PRISMA checklist”.

Additional Editor Comments (if provided):

Dear authors,

two independent reviewers have now revised your submission.

They agree in highlighting several merits of your review and meta-analysis.

Nonetheless, they also raise one major issue related to the theoretical framework adopted and they additionaly suggest minor edits that may improve your manuscript.

Thus, please submite a revised version of your manuscript alongside a point-by-point response to the reviewers' comments.

Sincerely,

Livio Provenzi

Reviewers' comments:

Reviewer's Responses to Questions

**Comments to the Author**

1. Is the manuscript technically sound, and do the data support the conclusions?

Reviewer #1: Yes

Reviewer #2: Yes

2. Has the statistical analysis been performed appropriately and rigorously? 

Reviewer #1: Yes

Reviewer #2: Yes

3. Have the authors made all data underlying the findings in their manuscript fully available?

Reviewer #1: Yes

Reviewer #2: Yes

4. Is the manuscript presented in an intelligible fashion and written in standard English?

Reviewer #1: Yes

Reviewer #2: Yes

5. Review Comments to the Author

Reviewer #1: This is a very well-written manuscript describing a meta-analysis examining the relation between interparental relationship quality (IPR) and sibling relationship quality (SRQ). Their aims were clear and I liked that they couched the work within a family systems perspective and put two competing hypotheses out there that could, and have been, used to explain the relation between IPR and SRQ: the spillover or compensatory hypothesis. There has been some speculation for years that a positive sibling relationship can buffer the effects of interparental conflict, but I don’t believe anyone until now has examined this assumption in a meta-analysis so the information here is timely and could make a contribution to the field.

I very much enjoyed reading this paper and found it quite informative.

I have some very minor comments for the authors that might help clarify some of their procedures and decision-making when conducting the meta-analysis.

Given the central focus on IPR, I think it is important from the start to define or describe what they are including here under the IPR category. It may seem obvious to some, but I wasn’t clear what the inclusion criteria were here. For instance, martial relationships clearly would be included, and one of their moderator variables was cohabiting versus non-cohabiting, so there was some consideration of residence. Many would consider the coparenting relationship to be perhaps a central feature of IPR but it appears that the coparenting literature was not examined here or included, so the question I have is why not? Why would the coparenting literature not be included as part of a meta-analysis on IPR? Perhaps there are not many studies out there, which I could see might be the case. But, some justification I believe needs to be included because this was a glaring hole in my read of this paper. So noting up front how they are defining IPR and what is included seems essential.

Also, on p. 3 when introducing the three aims, perhaps they can just list in parentheses for aim 3, what some of the moderators are that will be included to inform the reader of what is to come. I found myself asking. What moderators?

I understand that certain decisions have to be made when conducting meta-analyses, but I also think there needs to be some note then in the limitations section of the implications of such decisions and how the findings might have differed. I’m thinking specifically of the decision to use data from the younger sibling for the sake of simplicity. I agree that such decisions need to be made, but the literature is also pretty clear that older siblings are more likely to be the leaders, managers, and teachers in sibling relationships, and one might argue protectors or instigators, so perhaps IPR would have a much stronger effect when examining the older siblings than the younger ones. Often times, it is what the older sibling is doing that determines the SRQ. Perhaps the authors would have found stronger effects for compensation if the decision was to extract information on the older sibling versus the younger sibling.

As a follow-up, I was a bit confused later on p. 11 when they claimed they coded sibling order of the index child to include whether they were the younger or youngest sibling or the older or oldest sibling. This seemed to contradict their earlier decision to extract data only from younger siblings. Some clarification here might be helpful.

I was also not clear on what search terms were actually used. I can understand that a large number of hits could be found using terms such as “child”, “maternal”, etc. But some information on the iterative process of refinement that was used would be helpful here as I was not clear what “broad terms not specific to dyadic relationship descriptions” were.

P. 13 ‘…studies were excluded because abstract analysis revealed they were unsuitable for the current review.” Again, could the authors simply provide an example or two here of what these situations were that led to exclusion.

Reviewer #2: Dear editor, thank you for the opportunity to revise the manuscript “A systematic review and meta-analysis of the associations between interparental and sibling relationships: Spillover or compensation?”. This is a very relevant study and has many strengths. The statistical analysis and the study description are very accurate, and authors provided all details and materials to allow study replicability. However, I think that there are major and minor issues that should be addressed. Here are some comments and suggestions that may help authors to further improve their manuscript:

Major point

• My main concern is about the choice and presentation of the theoretical framework (i.e. framework by Erel and Burman and spillover vs. compensation hypothesis). It seems to me that a more consistent replication of the framework by Erel and Burman about siblings’ relationship would be a meta-analysis about the link between parent-child relationship and siblings’ relationship, where the feelings experienced by a child in the relation with a parent may spill over in her/his relationship with the sibling. I agree with authors about the presence of a number of reasons to expect that positive interparental relationship quality is associated with positive sibling relationship quality, and vice versa, however not all of them would be explained through the spillover mechanism: spillover mainly refers to an indirect impact on other family members through affect spillover within a person from one family subsystem to another one (i.e. it is an indirect hypothesis about the impact of interparental conflict on child adjustment which is explained trough the deteriorating of parenting practices). Alternative hypotheses to the spillover one, support a direct impact of interparental conflict on child adjustment (e.g. the emotional security hypothesis and the social learning theory), however they are reported in this manuscript in support the spillover mechanism. Also the influence of third family stressors (p. 6 line 130) is reported here as a form of spillover, however it is better known in literature as “crossover effect” (e.g. doi: 10.1037/a0015977 “A third hypothesized process is crossover. Rather than a transfer of affect within one person across subsystems (i.e. spillover), crossover refers to the transfer of affect or behavior between people. An example of crossover is when the stress experienced by one partner at work is detrimental to the other partner’s relationship with a child”). I think that these theories refer to different mechanisms (though they are not self-excluding) and should be presented in a more consistent way in the manuscript.

Minor points:

• I suggest maintaining consistency throughout the paper in the presentation of goals to help the readers (they are presented as 2 or 3 goals in different manuscript’s sections).

• Did the author address the potential overlap between samples of studies by the same author (e.g. Stocker, Tucker, Ruff, Brody)?

• Please, deepen also theoretical implications of your results in the discussion section.

6. PLOS authors have the option to publish the peer review history of their article (what does this mean?). If published, this will include your full peer review and any attached files.

Reviewer #1: No

Reviewer #2: No

---

## [Author Response · Author response to Decision Letter 0]

9 Jul 2021

We have uploaded a reponse letter that responds to each point raised by the academic editor and the reviewers as a separate file labeled 'Response to Reviewers'.

---

## [Decision Letter · Decision Letter 1]

27 Jul 2021

PONE-D-21-07364R1

A systematic review and meta-analysis of the associations between interparental and sibling relationships: Positive or negative?

PLOS ONE

Dear Dr. Zemp,

Thank you for submitting your manuscript to PLOS ONE. After careful consideration, we feel that it has merit but does not fully meet PLOS ONE’s publication criteria as it currently stands. Therefore, we invite you to submit a revised version of the manuscript that addresses the points raised during the review process.

We look forward to receiving your revised manuscript.

Kind regards,

Livio Provenzi

Academic Editor

PLOS ONE

Journal Requirements:

Additional Editor Comments (if provided):

Reviewers' comments:

Reviewer's Responses to Questions

**Comments to the Author**

1. If the authors have adequately addressed your comments raised in a previous round of review and you feel that this manuscript is now acceptable for publication, you may indicate that here to bypass the “Comments to the Author” section, enter your conflict of interest statement in the “Confidential to Editor” section, and submit your "Accept" recommendation.

Reviewer #1: (No Response)

Reviewer #2: All comments have been addressed

2. Is the manuscript technically sound, and do the data support the conclusions?

Reviewer #1: Yes

Reviewer #2: Yes

3. Has the statistical analysis been performed appropriately and rigorously? 

Reviewer #1: Yes

Reviewer #2: Yes

4. Have the authors made all data underlying the findings in their manuscript fully available?

Reviewer #1: Yes

Reviewer #2: Yes

5. Is the manuscript presented in an intelligible fashion and written in standard English?

Reviewer #1: Yes

Reviewer #2: Yes

6. Review Comments to the Author

Reviewer #1: Thank you once again for letting me review this manuscript on the link between IPR and sibling relationship quality (SRQ). I appreciated the recognition that there may be different processes or mechanisms that might be responsible for any positive or negative associations between the two relationships, besides spillover or compensation. I did wonder in reading this version if the attempt to be so precise and restrict the spillover hypothesis to the affect contagion or behavior of one individual in the relationship to that same individual in a second relationship was so restrictive now, that some of the significance of the current analysis has been diminished.

I appreciate the clearer definition of IPR (the romantic, intimate aspect of the adult-adult relationship), and making it clear that they did not include coparenting. The problem however, comes back in the discussion when they want to interpret their results and keep referring back to parenting as a possible mechanism that explains the associations or mediates the IPR-SRQ associations. I think there needs to be some recognition in their discussion that the results may have been different if coparenting was the interparental relationship variable they used, and might even be a better test of spillover as it may very well include the parenting (or coparenting) mechanisms they would like to claim mediate the positive or negative associations. It is not clear from the current presentation why intimacy or conflict between adult romantic partners would be linked to SRQ,

I still believe there should be some mention in their discussion that results may have been affected by their decision to use information on the younger siblings when there were two or more siblings for the reasons I cited the first time (older siblings as leaders) or that they classified studies by the lower end of the age range. Making these decisions does not undermine the quality of the analysis done here or reduce the contribution it can make, but the limitations of one’s scholarly decisions should also be acknowledged.

Please specify by listing an example or two of what qualified as at-risk samples or clinical samples (bottom of p. 15).

I found the referral to moderators using letters (e.g., Moderator N) a bit frustrating as it meant I had to refer back to the table or text repeatedly to be reminded of what they were actually testing. Also, I could not figure out what moderator M was for some time, until I went back to the Intro as it is missing in the table and only referred to in their table note as Moderator M and only described in the text as Moderator M. I’m wondering if there might be some way to add some more descriptive information in the text as to what is actually being tested with moderators than relying solely on a letter descriptor.

I believe the discussion still needs some work. I understand they were criticized the first time by not being precise on what spillover was and appear to have now restricted it to relationships with the same individual, although I’m not sure everyone would agree with this strict definition. The affective arousal and emotion dysregulation children experience when witnessing parents argue in an emotionally charged conflict could very well carry over into interactions between siblings, and not be mediated by parenting. I think the work of Mark Cummings demonstrated this years ago.

“Another theoretical implication of this study relates to our impression that scholars often use different labels and operational definitions for similar constructs in this field. Inconsistence in the terminology and confusion in the conceptual operationalization of family relationships constitute a major challenge particularly for meta-analyses. We tried to avoid potential pitfalls by a prudent coding of the operational definitions of the different dimensions of relationship quality. However, the field would generally benefit from greater conceptual consensus.” (p. 26).

This section requires some more elaboration as to what point they are trying to communicate. Is the idea here that some researchers are measuring different constructs (e.g., some study aggression, others conflict or antagonism), or that they are using different means (observations, parent report) to assess the same construct (aggression)? I’m not seeing how this is a major problem, other than perhaps in trying to classify for a meta-analysis, as wouldn’t we have stronger evidence if the effect was there across multiple methods? And if it is such a major problem, then perhaps the authors may want to make some recommendations for how to remedy this in future work, other than just noting it is a problem.

I also didn’t follow their logic for why it was plausible for there to be more spillover processes in families with girls based on gender socialization. Stating it as such and explaining it are two different means of discussing the results.

I’m not sure their findings support the recommendations for a focus on the IPR for intervention. They were clear they were focusing on the intimate adult romantic nature of IPR but all their recommendations are about how parents should be provided the necessary skills to promote healthy sibling relationships. There seems to be a disconnect between what they did in their analyses and what they want to conclude from those analyses. Managing sibling conflict is a parenting (or coparenting) strategy, and not part of the intimacy of an adult romantic relationship. A couple-oriented relationship intervention to prevent child behavior problems via parenting would probably be a coparenting-intervention and there are some very successful coparenting interventions, but again, coparenting was excluded in the current meta-analyses.

I also think they need to be careful about making some overgeneralizations of the scarcity of programs focused on sibling aggression and conflict. Although there may be few, they do exist, and there are certainly many parent-focused interventions to reduce children’s disruptive behavior, which could certainly be applied here in the sibling context. I liked their example of the Triple P program.

Reviewer #2: Authors carefully addressed all my previous comments and have modified the paper accordingly. I think that they much improved their paper that is now more consistent also from a theoretical point of view. It is a high quality piece of work, congratulations!

7. PLOS authors have the option to publish the peer review history of their article (what does this mean?). If published, this will include your full peer review and any attached files.

Reviewer #1: No

Reviewer #2: **Yes: **Serena Grumi

---

## [Author Response · Author response to Decision Letter 1]

1 Sep 2021

Dear Editor, 

Dear Reviewers, 

We would like to thank the editor and the two anonymous reviewers again for the helpful comments about our manuscript. The first Reviewer raised further important points that we have considered in our second revision. We responded to each point and made changes to the manuscript (see file labeled 'Revised Manuscript with Track Changes'). As requested, we also submit an unmarked version of the manuscript without tracked changes (see file labeled 'Manuscript'). The page numbers we provided in this response letter refer to the second, unmarked version of the manuscript. 

To comprehensively respond to all reviewer feedback, we pasted their comments into this document. Our responses are noted with Author Response in bold type and we note the location (pages) of any changes made to the manuscript. 

Once again, thank you very much for the time and attention you devoted to this manuscript. We believe these revisions resulted in a better quality piece of work.

Sincerely, 

Authors

 

Reviewer #1: 

Thank you once again for letting me review this manuscript on the link between IPR and sibling relationship quality (SRQ). I appreciated the recognition that there may be different processes or mechanisms that might be responsible for any positive or negative associations between the two relationships, besides spillover or compensation. I did wonder in reading this version if the attempt to be so precise and restrict the spillover hypothesis to the affect contagion or behavior of one individual in the relationship to that same individual in a second relationship was so restrictive now, that some of the significance of the current analysis has been diminished.

Author Response: Thank you for providing us constructive and detailed feedback a second time. We respond to each comment hereafter, including the point about the theoretical restrictions we made in the first revision (in particular, see our response to Comment 5). We hope that you agree that our manuscript improved again by these new revisions.

Comment 1: I appreciate the clearer definition of IPR (the romantic, intimate aspect of the adult-adult relationship), and making it clear that they did not include coparenting. The problem however, comes back in the discussion when they want to interpret their results and keep referring back to parenting as a possible mechanism that explains the associations or mediates the IPR-SRQ associations. I think there needs to be some recognition in their discussion that the results may have been different if coparenting was the interparental relationship variable they used, and might even be a better test of spillover as it may very well include the parenting (or coparenting) mechanisms they would like to claim mediate the positive or negative associations. It is not clear from the current presentation why intimacy or conflict between adult romantic partners would be linked to SRQ.

Author Response: Thank you for bringing this issue to our attention again. We tried to enhance clarity of the discussion in the matter of (co)parenting as a theoretical mechanism and stated more clearly that it is a study limitation that we have not tested it empirically. Please see our revisions on page 25:

“On the other hand, according to the spillover hypothesis, the interparental relationship can influence the sibling relationship indirectly through parenting or coparenting, respectively [3,17]. For instance, interparental conflict can disrupt childrearing practices or interfere with sensitive parenting [18]. A meta-analysis found a significant association between interparental conflict and dysfunctional parenting behaviors, with strongest effect sizes regarding harsh discipline and parental acceptance [19].” 

[…] 

“This means that the present data, as (co)parenting was not examined, do not reveal whether spillover of affect within parents underlies our findings. This is an important limitation and we may assume that the association between coparental alliance, parenting behavior, or parent-child interactions and sibling relationship quality would have yielded stronger effects than the association between interparental and sibling relationships. Future research needs to examine those postulated mechanisms.”

Comment 2: I still believe there should be some mention in their discussion that results may have been affected by their decision to use information on the younger siblings when there were two or more siblings for the reasons I cited the first time (older siblings as leaders) or that they classified studies by the lower end of the age range. Making these decisions does not undermine the quality of the analysis done here or reduce the contribution it can make, but the limitations of one’s scholarly decisions should also be acknowledged.

Author Response: Thanks for pointing this out again. We recognize that our decision to focus on the younger sibling merits mention in the limitations section. We added it on page 32:

“Last, for reasons of simplicity, only data from the younger (or youngest) child were extracted if characteristics of two (or more) siblings were reported. This decision has some disadvantages, as older siblings are more likely to be the leaders, superiors, and protectors in the sibling dyad. Hence, we cannot preclude the possibility that the pattern of results would have looked different when examining the older siblings’ perception of the sibling relationship quality.”

Comment 3: Please specify by listing an example or two of what qualified as at-risk samples or clinical samples (bottom of p. 15).

Author Response: We think that our criteria to code samples as at-risk or clinical are clearly described on page 12: 

“Studies with community samples do not focus on a particular subpopulation, thus, are based on non-stressed, healthy subjects. At-risk samples report elevated levels of stress or, respectively, deal with stressful life circumstances, such as below-average household income, parental unemployment, chronic illness of a family member, birth of a child with a physical disability, or presence of interparental violence. Clinical samples differ from at-risk samples in the level of psychopathology. In the present meta-analysis, studies are categorized as clinical samples if they either recruited participants in a clinical context (e.g., outpatient clinic, psychiatric consultation) or if any family member (parent, sibling) were diagnosed with a mental disorder.”

However, we agree with the Reviewer’s comment. We therefore added concrete examples of the primary studies concerning the categorization of the sample type on pages 15/16:

“With k = 42 (89%), most studies referred to community samples (healthy subjects without known risk), while k = 4 (9%) investigated at-risk samples (e.g., low-income families, families exposed to community violence, families with intimate partner violence). One study (2%) examined all three population categories – community, at-risk, and clinical – separately in subgroups within the same investigation (i.e., children diagnosed with current major depression with a depressed parent, depressed children without a depressed parent, children considered at high-risk for depression, and children considered at low-risk for depression (normal controls); see Weaver-Graham [77]).”

Comment 4: I found the referral to moderators using letters (e.g., Moderator N) a bit frustrating as it meant I had to refer back to the table or text repeatedly to be reminded of what they were actually testing. Also, I could not figure out what moderator M was for some time, until I went back to the Intro as it is missing in the table and only referred to in their table note as Moderator M and only described in the text as Moderator M. I’m wondering if there might be some way to add some more descriptive information in the text as to what is actually being tested with moderators than relying solely on a letter descriptor.

Author Response: This is an important point concerning reader-friendly presentation. In the results section, we now specified each moderator letter by adding its description (particularly with regard to moderator [M]). It should now be possible to understand the results without referring back to the methods section.

Comment 5: I believe the discussion still needs some work. I understand they were criticized the first time by not being precise on what spillover was and appear to have now restricted it to relationships with the same individual, although I’m not sure everyone would agree with this strict definition. The affective arousal and emotion dysregulation children experience when witnessing parents argue in an emotionally charged conflict could very well carry over into interactions between siblings, and not be mediated by parenting. I think the work of Mark Cummings demonstrated this years ago.

Author Response: Thanks for suggesting to emphasize more the potential direct association, namely not mediated by parenting. This is an important piece of the puzzle that has been missing in the earlier discussion. We revised this part of the discussion by citing, among others, Cummings’ early seminal work in this field (see pages 26/27): 

“However, it can be assumed that affect contagion also occurs from one family relationship to the other if no person is involved in both relationships. For example, as the experience of seeing or hearing displays of anger between parents is itself aversive to children, repeated exposure to interparental hostility takes a direct toll on children. When faced with interparental conflict, children react emotionally (e.g., fear, distress), behaviorally (e.g., intervening, avoidance), physiologically (e.g., skin conductance level reactivity), and cognitively (e.g., insecure representations of parental relationship, self-blaming) [89]. Children’s reactivity to marital conflict is, according to the EST [6], an expression of perceived threat to their sense of security. Early experimental studies have demonstrated that children’s exposure to adult background anger, i.e. an angry interaction of two actors in the background of the experimental room, increased their behavioral and emotional distress and heightened aggression toward peers [90,91]. Similarly, exposure to videotaped couple arguments increased the likelihood of aggressive behavior in children [92]. From that perspective, a direct association between interparental conflict and sibling aggressive interactions, not mediated by parenting, is very plausible.”

Comment 6: “Another theoretical implication of this study relates to our impression that scholars often use different labels and operational definitions for similar constructs in this field. Inconsistence in the terminology and confusion in the conceptual operationalization of family relationships constitute a major challenge particularly for meta-analyses. We tried to avoid potential pitfalls by a prudent coding of the operational definitions of the different dimensions of relationship quality. However, the field would generally benefit from greater conceptual consensus.” (p. 26).

This section requires some more elaboration as to what point they are trying to communicate. Is the idea here that some researchers are measuring different constructs (e.g., some study aggression, others conflict or antagonism), or that they are using different means (observations, parent report) to assess the same construct (aggression)? I’m not seeing how this is a major problem, other than perhaps in trying to classify for a meta-analysis, as wouldn’t we have stronger evidence if the effect was there across multiple methods? And if it is such a major problem, then perhaps the authors may want to make some recommendations for how to remedy this in future work, other than just noting it is a problem.

Author Response: Thank you, we revised this passage in the discussion. We do not regard the use of multiple methods as a problem. We agree with you that this is, on the contrary, an important strength to maximize the validity of relationship measures. Rather, with this comment in the discussion we tried to emphasize the inconsistent use of terminology for similar constructs. To clarify this issue, we added examples what we meant by this statement and closed the paragraph by a recommendation. On page 27, it now reads:

“Another theoretical implication of this study relates to our impression that scholars often use different labels and operational definitions for similar constructs in this field (e.g., relationship satisfaction / quality / adjustment / functioning for positive dimensions or conflict / distress / discord / disagreements for negative dimensions of relationship measures). Inconsistence in the terminology and confusion in the conceptual operationalization of family relationships constitute a major challenge particularly for meta-analyses, as it is rarely possible to create an exhaustive list of search terms. We tried to avoid potential pitfalls by a prudent coding of the operational definitions of the different dimensions of relationship quality. Although relationship science is complex and manifold and there are no established international definitions, the field would generally benefit from greater conceptual consensus. We recommend that empirical studies at least mention frequently used global key words in their articles (e.g., relationship quality) if they focus on one narrow indicator of it (e.g., intimacy).”

Comment 7: I also didn’t follow their logic for why it was plausible for there to be more spillover processes in families with girls based on gender socialization. Stating it as such and explaining it are two different means of discussing the results.

Author Response: We provided further explanations and previous findings with regard to this topic. We hope that our reflections became clearer in this revision, see pages 27/28:

“One potential explanation of this finding are developmental models of gender socialization: Girls are commonly socialized to value interdependence and connectedness in close relationships while boys are often supported to develop greater independence and autonomy [93]. As a result, boys are hypothesized to develop greater concern for self-protection, whereas girls experience pressure to conform to communal gender roles that are manifested in greater interest toward communion and the merging of individuals within social systems [94]. Prior research has found that they are particularly likely to ruminate about and intervene in family interactions, exhibit greater sensitivity to interparental disagreements, and report more self-blaming appraisals in the face of interparental conflict [95, 96]. Hence, gender differences are plausible in this meta-analysis, with girls being more likely to incorporate interpersonal behavior from their parents into their sibling interactions than boys. In other words, girls might be more susceptible to imitate relationship interactions leading to greater associations between the interparental and sibling relationship in families with daughters.”

Comment 8: I’m not sure their findings support the recommendations for a focus on the IPR for intervention. They were clear they were focusing on the intimate adult romantic nature of IPR but all their recommendations are about how parents should be provided the necessary skills to promote healthy sibling relationships. There seems to be a disconnect between what they did in their analyses and what they want to conclude from those analyses. Managing sibling conflict is a parenting (or coparenting) strategy, and not part of the intimacy of an adult romantic relationship. A couple-oriented relationship intervention to prevent child behavior problems via parenting would probably be a coparenting-intervention and there are some very successful coparenting interventions, but again, coparenting was excluded in the current meta-analyses.

Author Response: We have now tailored our practical implications to underline the importance of equipping parents with skills to model a positive interparental relationship for their children in order to promote healthy sibling relationships. While we agree that coparenting-focused interventions may be successful towards reducing child behavioral problems, we have decided to draw on evidence explicitly showing how strengthening the interparental relationship may lead to improved child behavior given the focus of this meta-analysis (as coparenting was not assessed). We thereby hope to eliminate the disconnect that you have pointed out. On pages 29/30, we revised the following passages:

“Parents act as central models of social relationships within the family. Supporting parents in strengthening their interpersonal bond can help them set a positive example and equip their children with the necessary skills to foster their own positive relationships. If we assume that the interparental relationship spills over into the sibling relationship, one route that could target this, would be to primarily strengthen the interparental relationship. A randomized controlled trial has shown that a couple-focused intervention can reduce child behavioral problems, mediated by enhanced relationship quality in mothers and improved parenting behavior in fathers [98]. While the importance of the enhancement of the interparental relationship to reduce and prevent child behavior problems is increasingly recognized [99], the effects of couple-oriented programs on the sibling relationship have been largely neglected so far.”

[…]

“While targeting either the interparental or sibling domain may help improve family relationships, it may be worth for future work to embrace a more integrated family systems approach that holds promise to influence multiple family subsystems simultaneously. Couple-focused interventions including both components of interparental bonds and parenting or coparenting have been successful at reducing child behavior problems by improving the interparental relationship quality first and in turn the parent-child relationship [103].”

Comment 9: I also think they need to be careful about making some overgeneralizations of the scarcity of programs focused on sibling aggression and conflict. Although there may be few, they do exist, and there are certainly many parent-focused interventions to reduce children’s disruptive behavior, which could certainly be applied here in the sibling context. I liked their example of the Triple P program.

Author Response: Thank you for your comment. We certainly agree that it is important to be cautious about making overgeneralizations of the scarcity of programs tailored towards sibling aggression or conflict. However, we believe that we have made it clear that they do exist, but added two more references to emphasize this issue better, see page 30: 

“Programs that address sibling conflict and aggression are scarce, however there is a handful of studies that have put forward interventions that are aimed at either improving children’s social skills or at teaching parents mediation techniques in the face of sibling conflict [100-102].”

Moreover, as you commented, our reference to the Triple P parenting-program was intended to point out that there are parent-focused interventions that may be applied in order to improve sibling relationships, see pages 30/31: 

“Parenting-oriented programs, such as Triple P [104], on the other hand may hold promise to indirectly influence both the interparental and the sibling domain. Previous studies have found that variants of the Triple P training have the potential to enhance the interparental relationship quality [105], as well as to reduce sibling conflict [106].”

 

Reviewer #2: 

Authors carefully addressed all my previous comments and have modified the paper accordingly. I think that they much improved their paper that is now more consistent also from a theoretical point of view. It is a high quality piece of work, congratulations!

Author Response: Many thanks! We appreciate the acknowledgement of our revisions.

---

## [Decision Letter · Decision Letter 2]

14 Sep 2021

A systematic review and meta-analysis of the associations between interparental and sibling relationships: Positive or negative?

PONE-D-21-07364R2

Dear Dr. Zemp,

We’re pleased to inform you that your manuscript has been judged scientifically suitable for publication and will be formally accepted for publication once it meets all outstanding technical requirements.

Kind regards,

Livio Provenzi

Academic Editor

PLOS ONE

Additional Editor Comments (optional):

Reviewers' comments:

Reviewer's Responses to Questions

**Comments to the Author**

1. If the authors have adequately addressed your comments raised in a previous round of review and you feel that this manuscript is now acceptable for publication, you may indicate that here to bypass the “Comments to the Author” section, enter your conflict of interest statement in the “Confidential to Editor” section, and submit your "Accept" recommendation.

Reviewer #1: All comments have been addressed

2. Is the manuscript technically sound, and do the data support the conclusions?

Reviewer #1: Yes

3. Has the statistical analysis been performed appropriately and rigorously? 

Reviewer #1: Yes

4. Have the authors made all data underlying the findings in their manuscript fully available?

Reviewer #1: Yes

5. Is the manuscript presented in an intelligible fashion and written in standard English?

Reviewer #1: Yes

6. Review Comments to the Author

Reviewer #1: The authors have addressed all my comments and should be commended for conducting a very thorough and thoughtful meta-analysis on the links between IPR and sibling relationships.

7. PLOS authors have the option to publish the peer review history of their article (what does this mean?). If published, this will include your full peer review and any attached files.

Reviewer #1: No

---

## [Editor Report · Acceptance letter]

17 Sep 2021

PONE-D-21-07364R2 

A systematic review and meta-analysis of the associations between interparental and sibling relationships: Positive or negative? 

Dear Dr. Zemp:

I'm pleased to inform you that your manuscript has been deemed suitable for publication in PLOS ONE. Congratulations! Your manuscript is now with our production department. 

Kind regards, 

on behalf of

Dr. Livio Provenzi 

Academic Editor

PLOS ONE